# Destination Therapy Strategies of Advanced Heart Failure in Elderly Non-Heart Transplant Candidates: A Propensity Matching Analysis from the LEVO-D and REGALAD Registries

**DOI:** 10.3390/life14121570

**Published:** 2024-11-29

**Authors:** David Dobarro, Sergio Raposeiras-Roubin, Luis Almenar-Bonet, Eduard Solé-González, Mireia Padilla-Lopez, Carles Diez-Lopez, Javier Castrodeza, Maria Dolores García-Cosío, Marta Cobo-Marcos, Javier Tobar, Pau Codina, Silvia Lopez-Fernandez, Francisco Pastor, Diego Rangel-Sousa, Eduardo Barge-Caballero, Beatriz Diaz-Molina, Alfredo Barrio-Rodriguez, Virginia Burgos-Palacios, Jesús Álvarez-García, Oscar González-Fernández, Andrés Grau-Sepulveda, José Manuel Garcia-Pinilla, Sonia Ruiz-Bustillo, Ana B. Mendez-Fernández, David Vaqueriza-Cubillo, Igor Sagasti-Aboitiz, Miguel Rodriguez-Santamarta, Ainara Lozano-Bahamonde, Ana Abecia, Inés Gómez-Otero, Raquel Marzoa, Eva González-Babarro, Manuel Gómez-Bueno, José Gonzalez-Costello

**Affiliations:** 1Hospital Álvaro Cunqueiro, Complexo Hospitalario Universitario de Vigo, IIS Galicia Sur, 36312 Vigo, Spain; sergio.raposeiras.roubin@sergas.es; 2Hospital Universitario La Fe, CIBERCV, 46126 Valencia, Spain; lualmenar@gmail.com; 3Hospital Clinic i Provincial, 08036 Barcelona, Spain; edsole@clinic.cat; 4Hospital de la Santa Creu i Sant Pau, IIB SANT PAU, 08025 Barcelona, Spain; mpadilla@sanptau.cat; 5Hospital Universitari de Bellvitge, IDIBELL, Universitat de Barcelona, CIBER-CV, Hospitalet del Llobregat, 08007 Barcelona, Spain; cdiez@bellvitgehospital.cat (C.D.-L.); jgcostello@hotmail.com (J.G.-C.); 6Hospital General Universitario Gregorio Marañón, 28007 Madrid, Spain; jcastrodeza5@gmail.com; 7Hospital 12 de Octubre Madrid, IMAS12, CIBERCV, 28041 Madrid, Spain; mariadolores.garcia-cosio@salud.madrid.org; 8Hospital Universitario Puerta de Hierro, IDIPHISA, 28222 Madrid, Spain; martacobomarcos@hotmail.com (M.C.-M.); mgomezbueno@secardiologia.es (M.G.-B.); 9Hospital Clínico Universitario de Valladolid, 47003 Valladolid, Spain; tobolobo_10@hotmail.com; 10Hospital Germans Trias i Pujol, 08916 Badalona, Spain; pau.codi@gmail.com; 11Hospital Universitario Virgen de las Nieves, IBSGranada, 18016 Granada, Spain; silvia.lopez.sspa@juntadeandalucia.es; 12Hospital Universitario Virgen de la Arrixaca, 30120 Murcia, Spain; franpastor79@hotmail.com; 13Hospital Universitario Virgen del Rocio, 41013 Sevilla, Spain; diegorangelsousa@gmail.com; 14Complexo Hospitalario Universitario de A Coruña, CIBERCV, 15006 A Coruña, Spain; eduardo.barge.caballero@sergas.es; 15Hospital Universitario Central de Asturias, 33011 Oviedo, Spain; beadimo@gmail.com; 16Complejo Asistencial de Salamanca, 37007 Salamanca, Spain; alfredobrarrior@gmail.com; 17Hospital Universitario Marqués de Valdecilla, 39008 Santander, Spain; virginia.burgos@scsalud.es; 18Hospital Universitario Ramón y Cajal, CIBERCV, 28034 Madrid, Spain; jalvarez82@gmail.com; 19Hospital Universitario La Paz, 28046 Madrid, Spain; ojog_1@hotmail.com; 20Hospital Universitario Son Espases, 07120 Palma de Mallorca, Spain; andres.grau@ssib.es; 21Hospital Universitario Virgen de la Victoria, IBIMA, 29010 Malaga, Spain; marlucale41@gmail.com; 22Hospital del Mar, 08003 Barcelona, Spain; 96896@parcdesalutmar.cat; 23Hospital Universitari Vall d’Hebron, 08035 Barcelona, Spain; ana.mendez@vhebron.net; 24Hospital Universitario Infanta Leonor, 28040 Madrid, Spain; davidvaq@hotmail.com; 25Hospital Universitario de Cruces, 48903 Bizkaia, Spain; igor.sagastiaboitiz@osakidetza.eus; 26Complejo Hospitalario Universitario de León, 24008 León, Spain; mrsantamarta@gmail.com; 27Hospital de Basurto, 48013 Bilbao, Spain; ainara.lozanobahamonde@osakidetza.eus; 28Hospital de Navarra, 31008 Pamplona, Spain; acabecia@hotmail.com; 29Complexo Hospitalario Universitario de Santiago, 15706 Santiago de Compostela, Spain; maria.ines.gomez.otero@sergas.es; 30Hospital Arquitecto Marcide, 15405 Ferrol, Spain; raquel.marzoa.rivas@sergas.es; 31Hospital de Montecelo, Complexo Hospitalario Universitario de Pontevedra, 36071 Pontevedra, Spain; egbabarro@hotmail.com

**Keywords:** advanced heart failure, LVAD, inotropes

## Abstract

Heart transplantation (HT) is the gold standard therapy for advanced heart failure (ADHF), and LVADs as destination therapy are an option in non-HT candidates. Most patients with ADHF never receive HT or an LVAD, so alternative strategies are needed. Intermittent levosimendan can reduce HF hospitalizations in ADHF patients in the short term. It is uncertain whether the results of the comparison of inotropes with older-generation LVADs would have the same outcomes in the current era of ADHF patients treated with levosimendan, who are less sick but older. In this paper, we compare the use of two therapeutic strategies for end-stage HF in patients who are not candidates for HT: repetitive intermittent levosimendan vs. LVAD as destination therapy. To do so, we compare two multicenter cohorts of real-life patients from Spain: the LEVO-D registry and the REGALAD registry. In total, 715 patients coming from the two registries were found: 403 from LEVO-D and 312 from REGALAD. Non-adjusted median survival was shorter for LEVO-D patients, with the benefit for the LVADs seen only after the first year of therapy. The survival advantage for the LVAD cohort was also true after analysis of the matched cohort but, as in the non-matched analysis, the survival benefit was mainly shown after one year of follow-up. We conclude that in elderly ADHF non-HT candidates, LVAD therapy offers significantly better long-term outcomes when compared to intermittent levosimendan; thus, it should be considered in carefully selected candidates. On the other hand, in poor LVAD candidates or highly comorbid patients, intermittent inotropic support with levosimendan could be a reasonable alternative to LVAD, as 1-year outcomes are similar.

## 1. Introduction

Heart transplantation (HT) is the gold standard therapy for advanced heart failure (ADHF) [1]. However, many patients at this stage have contraindications that preclude them from receiving a transplant, necessitating alternative treatments. In recent decades, durable left ventricular assist devices (LVADs) have emerged as a viable alternative for those deemed unsuitable for HT, particularly following the development of the latest generation of centrifugal continuous-flow and fully magnetically levitated devices [2]. Factors such as advanced age, impaired renal and right ventricular function, and other comorbidities are prevalent in ADHF patients and are common reasons for excluding them from LVAD therapy [3]. Consequently, the majority of ADHF patients, especially the elderly population, do not receive either HT or a LVAD. Therefore, alternative strategies due to their high symptom burden, poor quality of life, and frequent hospital admissions become necessary.

Intermittent ambulatory administration of levosimendan has demonstrated, in several small randomized clinical trials, the ability to reduce heart failure hospitalizations in the short term for ADHF patients [4,5]. Real-world data also suggest that regular, intermittent use of this drug as a last resource for end-stage heart failure patients who are not candidates for advanced therapies might decrease hospitalizations for heart failure [6]. Levosimendan’s advantage of ambulatory administration [7] over other intravenous inotropes is remarkable, and patients treated with this strategy are generally less severely ill compared to those treated with milrinone or dobutamine [8,9]. Therefore, it remains uncertain whether the outcomes of comparisons between inotropes and older-generation LVADs are applicable in the current era, characterized by older but less severely ill ADHF patients treated with levosimendan.

In this paper, we compare two therapeutic strategies for end-stage heart failure in patients who are not candidates for HT: repetitive intermittent levosimendan administration versus LVAD as destination therapy. This comparison has not been re-examined in recent years, despite changes in the patient profile for ADHF and advancements in LVAD technology. To achieve this, we analyze two multicenter cohorts of real-world patients from Spain: the LEVO-D Registry and the REGALAD Registry.

## 2. Materials and Methods

The LEVO-D registry [6] is a multicentre, retrospective study involving patients over 18 years old diagnosed with ADHF who are not candidates for HT or LVAD. Twenty-three tertiary hospitals in Spain participated in the registry, including patients who received at least one dose of ambulatory levosimendan between 1 January 2015 and 1 September 2020. Patients needed to be on optimal medical therapy (OMT) as determined by their treating physician. Patients with de novo heart failure or those who underwent any procedure that could improve prognosis or clinical outcomes (such as coronary revascularization, valve repair or replacement, cardiac resynchronization therapy [CRT] device implantation, or any other procedure deemed by the investigator to improve prognosis or quality of life) after the initiation of levosimendan were excluded from the registry. OMT was defined based on current guidelines and did not include sodium-glucose co-transporter type 2 (SGLT2) inhibitors, as the patients in this registry were enrolled prior to the introduction of SGLT2 inhibitors.

The REGALAD registry [10] is an observational, multicentre study that includes all long-term LVAD procedures performed in adults in Spain from 2007 to 31 December 2021. All Spanish hospitals performing long-term LVAD implantation participated in this registry. At each centre, a local physician and a surgeon were responsible for entering all LVAD procedures into the REGALAD registry. For the current analysis, only devices used as destination therapy were included. This included the following LVAD intracorporeal continuous-flow models for isolated left ventricular support: INCOR (Berlin Heart GmbH, Berlin, Germany), HeartMate II (Abbott, Chicago, IL, USA), and Jarvik 2000 (Jarvik Heart Inc., New York, NY, USA), which provide axial flow, and HeartWare HVAD (Medtronic, Minneapolis, MN, USA) and HeartMate 3 (Abbott, Chicago, IL, USA), the latest-generation pumps that provide centrifugal continuous flow.

### 2.1. Centers Involved

All centers participating in both registries had a specialized or ADHF unit. Twenty-three tertiary hospitals in Spain were involved in the LEVO-D registry, and twenty-two participated in the REGALAD registry. Twelve hospitals participated in both registries. All hospitals in the REGALAD registry had the capacity for HT or LVAD implantation, but only twelve of the hospitals that enrolled patients in the LEVO-D registry were performing at least one of these advanced therapies at the time the patients were enrolled.

### 2.2. LEVO-D Data Collection

Baseline data were collected on the day of the first dose of levosimendan, with blood pressure and heart rate measured before the drug was administered. Routine urgent laboratory data including blood count, renal function, or N-terminal pro-brain natriuretic peptide (NT-ProBNP) were obtained on the day of the first scheduled infusion. Additional laboratory parameters not typically obtained in an urgent blood sample were allowed if the sample was taken up to 21 days before the first levosimendan administration. Echocardiographic data were obtained as close to the first infusion as possible. Data were collected in an anonymous database and analyzed after approval by the regional ethics committee. Patients were followed according to their clinician’s judgment, with outcomes updated until June 2021.

### 2.3. REGALAD Data Collection

The REGALAD registry includes most variables from the IMACS (International Society for Heart and Lung Transplantation Registry for Mechanically Assisted Circulatory Support) and EUROMACS (European Registry for Patients with Mechanical Circulatory Support) registries, as well as additional variables considered pertinent by the registry founders. These variables include demographic, clinical, laboratory, echocardiographic, and hemodynamic characteristics, implantation data, and follow-up data at 3 months, 1 year, and annually. Adverse events associated with the device were specifically recorded. Heart failure severity at implantation was graded using INTERMACS (Interagency Registry for Mechanically Assisted Circulatory Support) scale [9]. Patients were followed until death or study closure on 31 December 2021.

### 2.4. Endpoints

We analyzed all-cause death during follow-up and the combined endpoint of death or heart failure hospitalizations at 1 year after receiving at least one dose of levosimendan or the LVAD implant.

### 2.5. Statistical Analysis

Results are expressed as mean ± standard deviation, medians (interquartile range [IQR]), or percentages, depending on the variable. Statistical differences were analyzed using Student’s *t*-test (for Gaussian distribution), nonparametric Mann–Whitney U test (non-Gaussian distribution), nonparametric McNemar test, or chi-squared test as appropriate. Survival analysis was performed using Kaplan–Meier curves, and the log-rank test was used to evaluate statistical significance. Missing data were managed by performing multiple imputations for all relevant parameters in the entire population. SPSS version 25 was used for multiple imputations using the automated function. A 20% limit for missing data was set to exclude variables with excessive missing data. Results were expressed as hazard ratio (HR) with a 95% confidence interval (95% CI). A *p*-value < 0.05 was considered significant.

### 2.6. Propensity Matching

Propensity score matching (PSM) was performed to minimize the bias inherent to observational studies. First, the propensity score was used to assess the probability of each patient being treated with levosimendan or LVAD, according to baseline characteristics. Subsequently, PSM was performed to match the characteristics of both groups (LEVO-D vs. REGALAD). We used a 1:1 protocol without replacement and calipers of width equal to 0.2 of the standard deviation of the logit of the PSM. All the measured covariates were well balanced across the comparator group (Figure 1).

PSM was conducted using binary logistic regression, with the dependent variable being the type of ADHF therapy (LVAD vs. levosimendan). The explanatory variables were age, sex, body mass index, hypertension, diabetes mellitus, atrial fibrillation/flutter, ischemic heart disease, NYHA class, INTERMACS, systolic blood pressure, systolic pulmonary arterial pressure, heart rate, LVEF, right ventricular dysfunction, moderate–severe mitral regurgitation, moderate–severe tricuspid regurgitation, ≥3 HF hospitalizations, and medical therapy (beta-blockers, angiotensin-converting-enzyme inhibitors/angiotensin receptor blockers -ACEI/ARB-, angiotensin receptor/neprilysin inhibitor -ARNI, mineralocorticoid receptor antagonist -MRA-, amiodarone, digoxin, hydralazine, thiazide, furosemide, oral anticoagulation).

## 3. Results

A total of 715 patients were identified from the two registries, comprising 403 from the LEVO-D registry and 312 from the REGALAD registry. Among the latter, 104 patients were classified as receiving an LVAD for destination therapy. This group included patients who received an LVAD as either destination therapy or as a bridge to candidacy, though they were deemed unlikely to undergo HT by their medical team and ultimately did not receive a heart transplant. The majority (91.3%) of patients in the REGALAD cohort were implanted with a third-generation continuous-flow centrifugal device (56.7% HeartMate 3 and 34.6% HeartWare). Among the remaining patients, one received a Jarvik 2000, five received a HeartMate II, and three received a Berlin Heart INCOR. Although the REGALAD database included patients from 2007 onward, 92.3% of the REGALAD patients had their implant performed between 2015 and 2020, and 95.1% between 2014 and 2020; thus, the majority of patients were contemporaneous with the LEVO-D registry. Full cohort characteristics of the LEVO-D and REGALAD registries have been published elsewhere [5,8].

### 3.1. Demographics

Table 1 presents a comparison of the main characteristics of the two groups. Patients referred for destination inotropes were slightly older. Left ventricular ejection fraction (LVEF) and pulmonary pressures were also higher in the LEVO-D group, and left ventricular end-diastolic diameter (LVEDD) was smaller. Furthermore, the LEVO-D group exhibited worse renal function, bilirubin levels, and natriuretic peptides, but a higher proportion of patients in the REGALAD cohort were in NYHA Class IV. More LEVO-D patients had atrial fibrillation or flutter and therefore more frequently received anticoagulation. More REGALAD patients were on amiodarone and had an implantable cardioverter-defibrillator (ICD) implanted. Regarding the neurohormonal blockade, differences were found only in beta-blockers; 65% of patients in the REGALAD cohort were on beta-blockers compared to 78.9% in the LEVO-D cohort.

### 3.2. Survival and HF Events

No patients were lost to follow-up. During the follow-up period, 52.3% of LEVO-D patients and 42.3% of REGALAD patients died. Consequently, a higher proportion of patients in the inotrope group died despite having a shorter follow-up duration (median: 458 days [95% CI: 412–519] vs. 494 days [95% CI: 357–730]; *p* = 0.08). The non-adjusted median survival was shorter for LEVO-D patients (741 days [95% CI: 611–870] vs. 1486 days [95% CI: 969–2002]; *p* = 0.03), with the survival benefit for LVAD patients becoming apparent after the first year of therapy, as illustrated by the Kaplan–Meier curves (Figure 2A).

Within the first year of follow-up, the number of heart failure (HF) admissions did not differ significantly between the two groups (LEVO-D: 1.12 ± 1.78 vs. REGALAD: 0.94 ± 1.64; *p* = 0.35). However, the percentage of patients alive without admissions was higher among LEVO-D patients during this period (LEVO-D: 49.6% vs. REGALAD: 33.7%; *p* = 0.004).

### 3.3. Propensity Score Matching

Two groups of 53 matched patients were created. The predictive power of the model used to generate the propensity score was 0.91, with adequate calibration (Hosmer–Lemeshow test, *p* = 0.292). The survival advantage for the LVAD cohort was also confirmed in the matched cohort analysis (Table 2). Notably, all patients included in the propensity score matching (PSM) from the REGALAD registry were implanted from 2015 onward, aligning them temporally with the LEVO-D patients.

LVAD patients demonstrated significantly better survival compared to the LEVO-D cohort (median survival in days: 1574 [95% CI: 1048–1923] vs. 612 [95% CI: 296–927]; *p* = 0.042). As in the non-matched analysis, the survival benefit for LVAD patients was primarily observed after one year of follow-up (Figure 2B). However, the apparent one-year advantage for the LEVO-D cohort in terms of the combined event of death and HF hospitalization was not consistent in the PSM. The percentage of patients alive without admission within the first year was similar between the groups (LEVO-D: 37.7% vs. REGALAD: 39.6%; *p* = 0.84).

## 4. Discussion

Heart transplantation (HT) is considered the gold standard for advanced heart failure (ADHF); however, in the Western world, many ADHF patients have formal contraindications for HT, primarily due to advanced age. Therefore, alternative treatments are needed to decrease symptoms, reduce hospital admissions, and improve the quality of life for patients with ADHF. Among the commonly utilized options for the ADHF patients without access to HT are ambulatory inotropes and left ventricular assist devices (LVADs) as destination therapy. However, this scenario has not been revisited in recent years, despite changes in the ADHF patient profile and LVAD technology. The data we present aim to shed light on this real-life clinical dilemma.

Levosimendan is a calcium sensitizer that enhances the sensibility of cardiac troponin C to calcium within the sarcomere, without needing to increase intracellular calcium levels, which explains its inotropic and lusotropic effects [11]. The clinical development of levosimendan has been evaluated over the past decades, primarily in the context of acute heart failure in hospitalized patients with low cardiac output syndrome, and has been compared with placebo and other inotropes. In the RUSSLAN trial, levosimendan was compared with placebo in patients with post-infarction heart failure, reducing the risk of clinical deterioration and all-cause mortality by 46% at 14 days and 33% at 180 days [11]. However, clinical improvement was only seen in patients not treated with beta-blockers when compared to dobutamine in the SURVIVE study [11]. Levosimendan was also studied within this clinical scenario in the REVIVE trial [11], which included 600 patients with acute heart failure and severe ventricular dysfunction. In this study, levosimendan compared with placebo did show improvement in symptoms at 5 days, reduction in NT-ProBNP levels, and shorter hospitalization duration, but these benefits were accompanied by an increase in adverse events such as hypotension, ventricular tachycardia, and atrial fibrillation. In these studies, an initial bolus infusion was used, which was associated with the development of these adverse events, especially hypotension in patients with systolic blood pressures below 100 mmHg, leading to the abandonment of this approach in favor of continuous infusion only.

In contrast to other inotropic agents, the peak effect of levosimendan occurs over some hours, but its active metabolite, OR-1896, has a prolonged half-life (7 to 10 days) and contributes to the drugs’ longer-lasting effects. This makes levosimendan an attractive option for several clinical scenarios, for example, when planning mechanical circulatory support weaning or for treating ADHF patients in the outpatient setting with intermittent doses of the drug, as it was used in the patients included in the LEVO-D registry. Repeated outpatient infusions of levosimendan could partially address the unmet therapeutic needs of patients with advanced heart failure, either because they are not candidates for other advanced therapies, as a bridge to such therapies, or even as a bridge to improvement associated with the gradual introduction of neurohormonal blockade treatment. Several trials exploring levosimendan effects in this scenario have been published so far, with conflicting results. Meanwhile, LevoREP and LeoDOR failed to show clinical improvements with levosimendan, whereas LAICA and LION-HEART, as well as several observational evidence, did. The LION-HEART study was a multicenter, randomized, controlled trial comparing levosimendan versus placebo with a 2:1 ratio in 69 patients with ADHF and reduced ejection fraction. The trial demonstrated a significant reduction in natriuretic peptide levels (primary outcome) and a reduction in the number of heart failure hospitalizations (secondary outcomes). The LION-HEART study implemented a protocol of 6 bi-weekly infusions of levosimendan administered on an outpatient basis at 0.2 mcg/kg/minute, without a bolus infusion, which also applied to most of the patients included in the LEVO-D registry [6].

LVAD emerged in recent decades as an option for bridge to HT, bridge to candidacy to HT, or even as destination therapy in non-HT candidates [12]. From early trials with first-generation devices, LVADs have shown significant improvements in survival [3] compared to medical treatment in patients with ADHF not eligible for HT. Modern devices have improved long-term survival in the last years when implanted with destination intention, from 58% with HeartMate II^®^ [13], 67.6% with HeartWare^®^ (Medtronic, Minneapolis, MN, USA) [14] to 73.2% in the case of HeartMate 3 [15]. The refinement of LVAD technology over the last two decades has not only contributed to improved survival but also reduced device-related complications such as pump thrombosis, stroke, and gastrointestinal bleeding. However, although current LVADs are seen as a better treatment than just medical management when HT is not an option, providing better survival but also quality of life [3], it should be noted that LVADs as destination therapy remain an option for the minority of patients with ADHF [2] as LVAD candidature requires a highly selected population [12,16]. Our analysis of these two contemporary multicenter real-life registries demonstrates how non-heart transplant candidates with outpatient ADHF are being referred for either an LVAD or ambulatory inotropes in a country with relatively high heart transplantation availability and an aging population. The vast majority of patients from the LEVO-D and REGALAD registries were contemporaneous (from 2015 onwards), which was true for all the patients included in the propensity score matching (PSM) analysis. Based on our findings, we conclude that patients with end-stage heart failure treated today with either an LVAD or intermittent inotropes as destination therapy have different clinical profiles, similar one-year outcomes, but long-term survival for LVAD patients in both unmatched and PSM analyses.

### 4.1. Patients Characteristics of the Full Cohort

Non-matched cohorts of LEVO-D and REGALAD, despite each patient having ADHF, revealed significant differences. Patients referred for levosimendan as destination therapy exhibited a complex clinical profile characterized by more comorbidities, deteriorating renal function, elevated pulmonary pressures, a higher prevalence of atrial arrhythmias, and a more diverse range of heart failure etiologies. In contrast, these patients demonstrated better left ventricular ejection fraction (LVEF), less mitral regurgitation, and less dilated left ventricles compared to those referred for LVAD therapy. It is likely that a high percentage of these patients had anatomic characteristics of the left ventricle (such as reduced LV cavity) and clinical characteristics, such as advanced renal failure with evidence of severe glomerular damage, that have been linked to poor outcomes with LVADs [16], which is why they were never considered as potential LVAD candidates.

While LEVO-D patients were generally older than their REGALAD counterparts, the age difference was not as pronounced as initially anticipated, with both groups predominantly in their seventies. The baseline characteristics strongly suggest that patients referred for LVAD therapy typically exhibited more pronounced low-output syndrome and fewer comorbid conditions, which theoretically made them ideal for being referred to mechanical circulatory support. This is further evidenced by the lower percentage of REGALAD patients on beta-blockers, which are often withdrawn in cases of low cardiac output. Additionally, the burden of ventricular arrhythmias was likely more significant among REGALAD patients, as evidenced by their greater reliance on antiarrhythmic medications such as amiodarone and the presence of implantable cardioverter defibrillators (ICDs), which is likely linked to the fact that REGALAD patients had worse LVEF. All these data suggest that REGALAD patients had more ADHF due to pure pump failure, while LEVO-D patients included individuals with a more heterogeneous presentation of heart failure and better LVEF, where clinical deterioration was not solely attributable to low output syndrome but also the progression of end-organ dysfunction, such as cardiorenal syndrome or pulmonary hypertension, reflecting the complexities of their conditions.

### 4.2. Long- and Short-Term Outcomes

When adjusted by PSM analysis, elderly patients in the REGALAD cohort had significantly better long-term survival. This suggests that every effort should be made to carefully identify patients within the group of ambulatory ADHF non-HT candidates who could benefit from an LVAD, even among elderly and comorbid patients. Although LVAD recipients older than 70 years show worse survival compared to younger patients, their quality-of-life improvement is at least as good, with superior functional capacity [17]. Therefore, the decision to implant an LVAD in the elderly should balance the increased risk of early mortality with a careful evaluation of the candidate. Age, as a component of the recent HeartMate 3 survival risk score [18], should be considered along with the usual multiparameter case evaluation and right heart failure prediction.

Our findings show that, while an LVAD is superior to inotropes in the long term, intermittent levosimendan could be comparable in terms of survival and heart failure events within the first year, even in the adjusted population. There were no differences in the rate of death or heart failure admissions within the first year. This should be taken into account when evaluating LVAD candidacy for elderly heart failure patients, particularly those with multiple non-cardiac comorbidities that may limit one-year survival and those at high risk of right heart failure after LVAD implantation. Levosimendan infusions have been linked to significant reductions in heart failure hospitalizations and increased quality of life in small, randomized trials and real-life registries [4,5,6], showing significant survival improvements compared to historical data in ADHF [14]. For these reasons, intermittent inotropic support with levosimendan has become a popular tool for ADHF patients in some regions, as it can be easily administered in the ambulatory setting following protocols like the LION HEART study [4]. This likely explains why patients referred for levosimendan as destination therapy are significantly less ill than those referred for continuous intravenous inotropes, with a lower threshold for levosimendan use in some European countries compared to continuous ambulatory infusions of other inotropes like milrinone, which have much higher one-year mortality rates [8].

### 4.3. Clinical Implications

Interpreting old clinical trials comparing medical treatment, including or excluding inotropes and LVADs, presents challenges for several reasons: contemporary patients have different one-year mortality rates with current optimal medical therapy [1], but LVAD technology, complications, and outcomes have also improved significantly [2]. Patients randomized to medical treatment in the REMATCH trial showed a one-year mortality of 75% [19], and it was 89% in the INTREPID trial [20], compared to 26.5% in the LEVO-D registry [6] or the 22% rate of death, LVAD implantation, or transplant in the RELEVANT-HF [21] and 34% in the MedaMACS registry [22]. A recent meta-analysis of ambulatory inotropic treatment suggested a one-month mortality rate of 4.2% [23], although studies included were heterogeneous and some were separated by more than 20 years. It should also be noted that LVADs are entirely different compared to medical treatments from 15 years ago. However, patients are older and more comorbid, even more so than those in recent pivotal trials: REGALAD patients were more than five years older than DT patients from the MOMENTUM 3 [24] and INTERMACS registries [25]. These factors likely contribute to the low rate of LVAD implantation in some countries [10], especially compared to the United States, where age is seen as an obstacle when referring patients for an LVAD, as it is related to poorer outcomes, especially in those over 75 years old [26]. However, when comparing adequately selected elderly patients, LVAD offers better long-term outcomes, and this must be considered when discussing therapeutic options for elderly ADHF patients.

Our work aims to provide insight into this situation by comparing contemporary ADHF patients, treatments, and technology to offer a realistic picture of today’s management of ambulatory ADHF non-HT candidates and what to expect from different therapeutic options. The ongoing SweVAD trial aims to compare LVADs to optimal medical therapy in patients with ADHF and may provide more clinical answers to the current management of these patients [27]. Furthermore, it also shows what to expect from ambulatory inotropes in potential LVAD candidates in regions with poor access to LVAD therapy.

### 4.4. Limitations

This is a retrospective study and is thus subject to inherent biases. Propensity score analysis is more robust than regression analysis but has certain weaknesses, as unmeasured confounding factors cannot be corrected. The relatively small number of patients in the REGALAD registry, due to the low number of LVAD implants in Spain, as well as the differences in the two populations, means the sample size for the propensity analysis was small. However, it should be noted that this is the only contemporary study comparing destination inotropes and LVADs. Patients were not matched by psychosocial factors, which can influence the decision against LVAD implantation without other medical contraindications. Although these factors have not been related to worse survival on support [28,29], they have been linked to an increased risk of complications and decreased quality of life.

## 5. Conclusions

In elderly patients with ADHF who are not candidates for HT, our results suggest that, with the caution of the retrospective nature of our work, LVAD therapy offers significantly better long-term outcomes compared to intermittent levosimendan. Therefore, LVAD therapy should be considered in selected candidates. Conversely, for patients who are poor candidates for an LVAD or have a high burden of comorbidities, intermittent inotropic support with levosimendan could be a reasonable alternative, as one-year outcomes are similar.

## Figures and Tables

**Figure 1 life-14-01570-f001:**
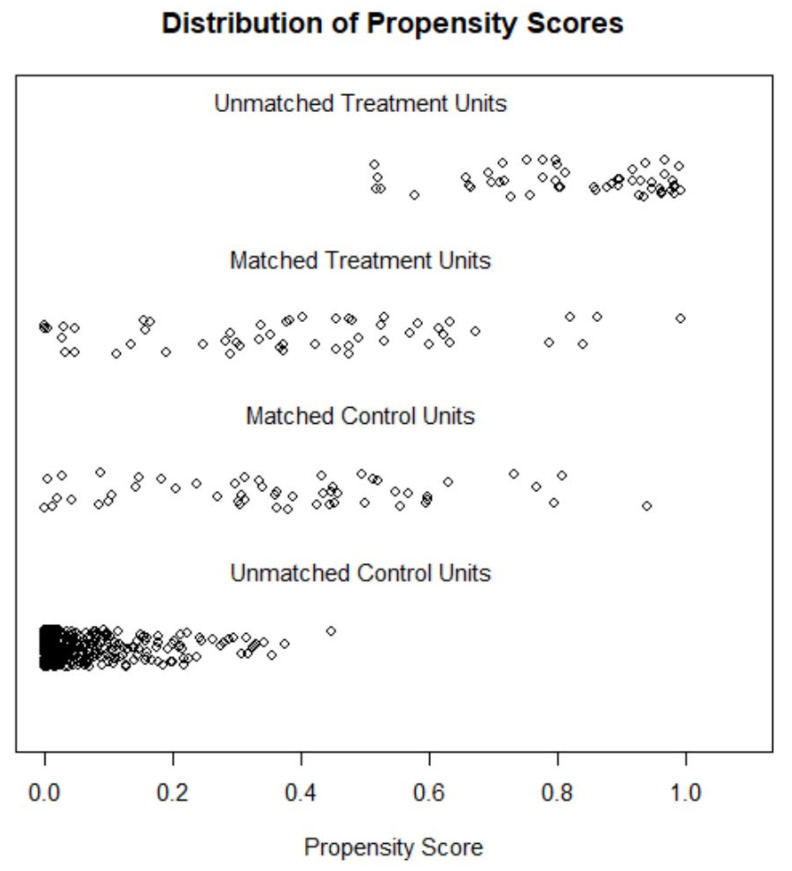
Distribution of propensity scores.

**Figure 2 life-14-01570-f002:**
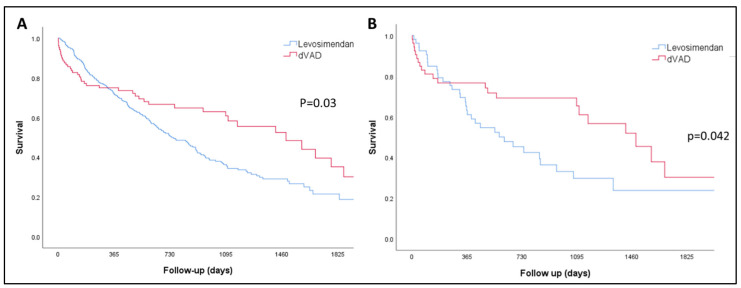
(**A**) Overall survival of the unmatched LEVO-D and REGALAD cohorts. (**B**) Overall survival of the matched LEVO-D and REGALAD cohorts. LVAD: left ventricular assist device.

**Table 1 life-14-01570-t001:** Demographics of the whole LEVO-D and REGALAD cohorts. BSA: body surface area. HF: heart failure. LVEF: left ventricular ejection fraction. LVEDD: left ventricular end-diastolic dimension. TAPSE: tricuspid annular plane systolic excursion. sPAP: systolic pulmonary artery pressure. RV: right ventricle. MR: mitral regurgitation. TR: tricuspid regurgitation. ACE-i: angiotensin-converting-enzyme inhibitors. ARB: angiotensin receptor antagonists. ARNI: angiotensin receptor neprilysin inhibitors. MRA: mineralocorticoid receptor antagonist. ICD: implantable cardioverter defibrillator. CRT: cardiac resynchronization therapy. NYHA: New York Heart Association. HR: heart rate. SBP: systolic blood pressure. DBP: diastolic blood pressure. NT-ProBNP: N-terminal pro-brain natriuretic peptide.

	LEVO-D	REGALAD	*p* Value
Age (years, mean)	69.4 ± 11.4	67.8 ± 5.7	0.046
Gender (male)	79.4%	80.8%	0.76
BSA (mean)	1.84 ± 0.2	1.85 ± 0.2	0.40
Hypertension	68.7%	67.3%	0.78
Diabetes mellitus	49.1%	46.2%	0.59
Atrial arrythmias	60.8%	30.8%	<0.001
Chronic kidney disease	54.1%	46.2%	0.15
HF diagnosis < 1 month	3.8%	4.8%	0.76
HF diagnosis 1–12 months	12.6%	11.5%	
HF diagnosis 1–2 years	9.8%	5.8%	
HF diagnosis > 2 years	73.3%	77.9%	
HF etiology DCM	26.8%	28.8%	0.002
HF etiology IHD	52.6%	65.4%	
HF etiology Others	20.6%	5.8%	
LVEF (%, mean)	27.5 ± 9.4	23.2 ± 6.2	<0.001
LVEDD (mm, mean)	63.2 ± 9.3	67.5 ± 10.2	<0.001
RV basal diameter (mm)	43.4	42.7	0.46
RV dysfunction (%)	67.5%	57.6%	0.54
TAPSE (mm, mean)	15.0 ± 4.3	16.0 ± 3.7	0.036
SPAP (mmHg, mean)	51.2 ± 15.8	47.5 ± 15.5	0.044
Severe MR	10.6%	22.1%	0.005
Severe TR	9.2%	5.8%	0.14
ACE-i/ARB/ARNI	70.0%	62.5%	0.14
Beta-blockers	78.9%	65.4%	0.004
MRA	69.7%	68.3%	0.78
Amiodarone	23.0%	36.5%	0.005
Anticoagulation	69.7%	54.4%	0.003
Digoxin	19.1%	16.7%	0.57
Furosemide or equivalent	96%	97.1%	0.60
ICD	55.1%	80.6%	<0.001
CRT	30.8%	36.7%	0.06
NYHA IV	12.9%	57.7%	<0.001
HR (bpm, mean)	73.3 ± 13.6	74.5 ± 12.8	0.27
SBP (mmHg, mean)	106.5 ± 15.5	102.5 ± 14.7	0.019
DBP (mmHg, mean)	63.4 ± 9.6	63.2 ± 9.9	0.83
HF admissions previous year	1.7 ± 1.6	2.0 ± 0.7	0.003
Sodium (mEq/L, mean)	138.6 ± 4.2	137.4 ± 4.5	0.018
Potassium (mEq/L, mean)	4.2 ± 0.6	4.1 ± 0.5	0.12
Urea (mg/dL, mean)	91.5 ± 51.2	78.8 ± 44.2	0.018
Creatinine (mg/dL, mean)	1.6 ± 0.7	1.4 ± 0.5	0.001
Bilirubin (mg/dL, mean)	1.2 ± 0.8	0.9 ± 0.5	0.002
Albumin (mg/dL, mean)	3.9 ± 0.5	3.8 ± 0.6	0.07
Uric acid (mg/dL, mean)	8.2 ± 2.8	7.9 ± 2.2	0.32
NT-ProBNP (pg/mL, mean)	9654 ± 11879	6203 ± 5995	<0.001
Hemoglobin (gr/dL, mean)	12.6 ± 1.9	12.0 ± 1.9	0.003
Lymphocytes (%)	19.8 ± 9.4	19.0 ± 7.8	0.44
HF admissions year after	1.12 ± 1.8	0.94 ± 1.64	0.35
Alive without admission 1 year after	49.6%	33.7%	0.004

**Table 2 life-14-01570-t002:** Demographics of the matched cohorts of the LEVO-D and REGALAD studies. BMI: body mass index. LVEF: left ventricular ejection fraction. LVEDD: left ventricular end-diastolic dimension. RV: right ventricle. MR: mitral regurgitation. TR: tricuspid regurgitation. sPAP: systolic pulmonary artery pressure. HR: heart rate. MDRD4. modification of diet in renal disease 4. NT-ProBNP: N-terminal pro-brain natriuretic peptide. ACE-i: angiotensin-converting-enzyme inhibitors. ARB: angiotensin receptor antagonists. ARNI: angiotensin receptor neprilysin inhibitors. MRA: mineralocorticoid receptor antagonist. ICD: implantable cardioverter defibrillator. CRT: cardiac resynchronization therapy. NYHA: New York Heart Association. OAC: oral anticoagulation.

	LEVO-D (53)	REGALAD (53)	*p* Value
Age (years, mean)	69.6 ± 8.9	68.4 ± 4.8	0.39
Sex (male)	73.6%	77.4%	0.87
BMI (mean)	25.8 ± 3.9	26.9 ± 3.8	0.16
Hypertension	64.2%	66%	0.84
Diabetes	52.8%	52.8%	0.99
Atrial tachyarrhythmias	39.6%	35.8%	0.84
IHD	60.4%	60.4%	0.99
LVEF (%, mean)	24.1%	23.5%	0.66
LVEDD (mm, mean)	66.5 ± 5.7	67.4 ± 10.8	0.60
RV dysfunction	54.7%	54.7%	0.99
MR III-IV	52.8%	56.6%	0.85
TR III-IV	37.7%	37.7%	0.99
SPAP (mmHg, mean)	49.2 ± 10.4	47.5 ± 13.3	0.47
SBP (mmHg, mean)	104.3 ± 16.6	103.2 ± 16.5	0.75
HR (bpm, mean)	75.9 ± 13.2	75.3 ± 12.7	0.81
MDRD4 (mL/min, mean)	53.7 ± 29.4	52 ± 23.6	0.67
NT-ProBNP (pg/mL, mean)	7910 ± 7000	7157 ± 6317	0.56
Hemoglobin (gr/dL, mean)	11.9 ± 1.7	11.9 ± 2.1	0.87
ACE-i/ARB/ARNI	62.2%	68%	0.68
Beta-blockers	67.9%	67.9%	0.99
ARM	69.7%	69.8%	0.99
Amiodarone	28.3%	30.2%	0.98
OAC	50.9%	56.6%	0.69
Digoxin	18.9%	20.8%	0.99
Furosemide	94.3%	94.3%	0.99
ICD	67.9%	71.7%	0.83
CRT	41.5%	47.2%	0.70
NYHA IV	41.5%	39.6%	0.99
3 or more HF admissions	20.8%	24.5%	0.82

## Data Availability

The original data are not publicly available. However, research projects are evaluated centrally by an ad hoc committee.

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
