# Peer review of "Destination Therapy Strategies of Advanced Heart Failure in Elderly Non-Heart Transplant Candidates: A Propensity Matching Analysis from the LEVO-D and REGALAD Registries"

_life, 2024, doi:10.3390/life14121570_

Round 1
Reviewer 1 Report
Comments and Suggestions for Authors
Your manuscript titled "Destination therapy strategies of advanced heart failure in elderly non-heart transplant candidates: A propensity matching analysis from the LEVO-D and REGALAD registries" is both compelling and thoroughly written. The research addresses a critical area in the management of advanced heart failure, particularly in elderly patients who are not candidates for heart transplantation. Your selection criteria and methodology are appropriate and well-detailed, which lends credibility and robustness to your findings.
To further enhance the comprehensiveness of your analysis, I would suggest incorporating more specific information on echocardiographic findings for patients in both the Levosimendan and LVAD groups. These additional details could provide a more in-depth understanding of the baseline characteristics and clinical profiles of each group, thereby enabling a more nuanced interpretation of the outcomes.
Additionally, a deeper exploration of the pharmacological effects of Levosimendan would be valuable. For example, discussing its potential impact on the hemodynamic and biochemical parameters observed could yield insights into the underlying mechanisms that might influence patient outcomes in this cohort.
Overall, the manuscript is of high quality, and these suggestions aim to further strengthen the clarity and depth of the findings.
Author Response
Dear reviewer,
Thank you very much for your review and constructive criticism. Your suggestions definetely makes this manuscript more robust and interesting. My response to your comments as follows:
Q1. "To further enhance the comprehensiveness of your analysis, I would suggest incorporating more specific information on echocardiographic findings for patients in both the Levosimendan and LVAD groups. These additional details could provide a more in-depth understanding of the baseline characteristics and clinical profiles of each group, thereby enabling a more nuanced interpretation of the outcomes"
In this version we have included the RV basal diameter as well as the percentage of patients with some degree of RV dysfunction in the table 1. This is added to the rest of echo parameters already included in the previous version of the manuscript (LVEF, LVEDD, TAPSE, SPAP and the degree of mitral and tricuspid regurgitation). Unfortunately, LEVO-D and REGALAD registries did not include any other echo information.
Q2. Additionally, a deeper exploration of the pharmacological effects of Levosimendan would be valuable. For example, discussing its potential impact on the hemodynamic and biochemical parameters observed could yield insights into the underlying mechanisms that might influence patient outcomes in this cohort.
We have included information about levosimendan pharmacology and main trials that explored its used in advanced heart failure, which includes its effect in biomarkers and haemodynamic variables (line 229 to line 260)
Overall, the manuscript is of high quality, and these suggestions aim to further strengthen the clarity and depth of the findings.
Thank you very much for this comment.
Reviewer 2 Report
Comments and Suggestions for Authors
This trial demonstrated that LVAD could be considered in selected elderly patients with ADHF who are not candidates for HT. Notably, it showed also that for poor candidates for LVAD or have a high burden of comorbidities, intermittent inotropic support with levosimendan could be a reasonable alternative
The trial looks enough original as it adds new useful data on arduous ADHF management.
The methodology is consistent with the aims, and so are the results and conclusions, which encourage other authors to address this issue.
The authors themselves correctly underline the trial's retrospective limitation. Moreover, propensity score analysis is more robust than regression analysis but has certain weaknesses, as unmeasured confounding factors cannot be corrected. However, it is the only contemporary study comparing destination inotropes and LVAD.
All the references are also appropriate and consistent with background and discussion.
Author Response
Dear reviewer,
Thank you very much for your kind comments about our work. We really celebrate that you found our work robust and interesting. This encourages us to keep going doing clinical research.
Reviewer 3 Report
Comments and Suggestions for Authors
Thank you for having the opportunity to read the article entitled "Destination therapy strategies of advanced heart failure in elderly non-heart transplant candidates: A propensity matching analysis from the LEVO-D and REGALAD registries." It is interesting because the authors talks about novel therapies in ADHF. However, I would suggest some revision in order to improve the quality of the manuscript:
1. Regarding the patient's characteristics, I would also include the chronic kidney disease since it is important in cardiovascular pathologies.
2. The authors mentioned in the discussion section that "Patients referred for levosimendan as destination therapy exhibited more comorbidities, worse renal function, higher pulmonary pressures, more atrial arrhythmias, and more heterogeneous heart failure etiology." I would suggest to comment more about this subject, about implication and pathophysiology, not just to repeat the results again in the discussion section.
3. The authors should comment more about those values that are statistically significant (p<0.001) and why they are different in the 2 groups (e.g. atrial arrhythmias, LVEF, ICD).
4. Regarding the discussion section, I would suggest to refer to more articles in the literature regarding those 2 therapies separately, not just the comparison between them (what other authors found on short and long-term outcome for just LVAD or just levosimendan).
5. What is the author's personal opinion regarding these results? Are there also other limitation regarding these findings for their worldwide use? For example, in some countries, the LVAD is not accessible. Please comment.
Comments on the Quality of English LanguageJust minor English editing (e.g. "for advanced heart failure (ADHF) and LVAD as destination therapy are an option in non-HT candidates"). Some sentences need to be revised for a better understanding.
Author Response
Dear reviewer,
Thank you very much for your review and constructive criticism. Your suggestions definetely makes this manuscript more robust and interesting. My response to your comments as follows:
Q1. Regarding the patient's characteristics, I would also include the chronic kidney disease since it is important in cardiovascular pathologies.
This version of the manuscript includes the percentage of patients with CKD (table 1)
2. The authors mentioned in the discussion section that "Patients referred for levosimendan as destination therapy exhibited more comorbidities, worse renal function, higher pulmonary pressures, more atrial arrhythmias, and more heterogeneous heart failure etiology." I would suggest to comment more about this subject, about implication and pathophysiology, not just to repeat the results again in the discussion section.
Following your suggestion, the section 4.1 of the discussion have been completely reformulated trying to better describe these differences and also trying to provide some insights to explain them (lines 280 to 312)
Q3. The authors should comment more about those values that are statistically significant (p<0.001) and why they are different in the 2 groups (e.g. atrial arrhythmias, LVEF, ICD).
Thank you for pointing out this. The new section 4.1 (lines 280 to 312) tries to better describe and explain these differences found in the non matched cohort.
Q4. Regarding the discussion section, I would suggest to refer to more articles in the literature regarding those 2 therapies separately, not just the comparison between them (what other authors found on short and long-term outcome for just LVAD or just levosimendan).
Thank you for suggesting this. The discussion section has been deeply revised and includes the information you kindly suggested (lines 229 to 271).
Q5. What is the author's personal opinion regarding these results? Are there also other limitation regarding these findings for their worldwide use? For example, in some countries, the LVAD is not accessible. Please comment.
What you suggest here is really interesting. We have included one sentence about this (lines 357-358).
QE. Comments on the Quality of English Language. Just minor English editing (e.g. "for advanced heart failure (ADHF) and LVAD as destination therapy are an option in non-HT candidates"). Some sentences need to be revised for a better understanding.
This version of the manuscript has been revised by a proffesional English translator thus we hope English have improved enough to meet journal´s high standards.
Round 2
Reviewer 3 Report
Comments and Suggestions for Authors
The authors responded to all if my questions. The article is suitable for publication.